# Associative detachment in anion-atom reactions involving a dipole-bound electron

Saba Zia Hassan [1], Jonas Tauch[1], Milaim Kas[2,3], Markus Nötzold[4], Henry López Carrera[1,5], Eric S. Endres[1,4], Roland Wester [4] & Matthias Weidemüller [1✉]

Associative electronic detachment (AED) between anions and neutral atoms leads to the detachment of the anion's electron resulting in the formation of a neutral molecule. It plays a key role in chemical reaction networks, like the interstellar medium, the Earth's ionosphere and biochemical processes. Here, a class of AED involving a closed-shell anion ($OH^-$) and alkali atoms (rubidium) is investigated by precisely controlling the fraction of electronically excited rubidium. Reaction with the ground state atom gives rise to a stable intermediate complex with an electron solely bound via dipolar forces. The stability of the complex is governed by the subtle interplay of diabatic and adiabatic couplings into the autodetachment manifold. The measured rate coefficients are in good agreement with ab initio calculations, revealing pronounced steric effects. For excited state rubidium, however, a lower reaction rate is observed, indicating dynamical stabilization processes suppressing the coupling into the autodetachment region. Our work provides a stringent test of ab initio calculations on anion-neutral collisions and constitutes a generic, conceptual framework for understanding electronic state dependent dynamics in AEDs.

[1] Physikalisches Institut, Ruprecht-Karls-Universität Heidelberg, 69120 Heidelberg, Germany. [2] Département de Chimie, Faculté des Sciences, Université Libre de Bruxelles (ULB), 1050 Brussels, Belgium. [3] Deutsches Elektronen-Synchrotron (DESY), 22607 Hamburg, Germany. [4] Institut für Ionenphysik und Angewandte Physik, Universität Innsbruck, 6020 Innsbruck, Austria. [5] Present address: Universidad de Las Fuerzas Armadas ESPE, 171103 Sangolquí, Ecuador. ✉email: weidemueller@uni-heidelberg.de

Anions are ubiquitous in nature, from aqueous solution[1] and the earth's atmosphere[2] to astrochemical environments[3–5]. They are reactive species, very sensitive to their environment, and often essential intermediates in important chemical events[6]. An important reactive process, distinguishing reactions involving anions from those involving cations or neutrals, is associative electronic detachment (AED), $A + B^- \rightarrow AB + e^-$, which leads to the formation of a neutral molecule. The reaction is energetically allowed if the dissociation energy of AB is greater than the electron affinity of B. Given their universality, investigations of AED have led to profound understanding of phenomena in diverse chemical reaction networks. In the interstellar medium, AED is assumed to be one of the main destructive mechanisms of astrochemically relevant anions[7]. It serves as an intermediate step in the creation of complex molecules[5,8], contributes to the production of molecular hydrogen[9] and the formation of interstellar water[10]. Furthermore, AED also plays a critical role in the formation of prebiotic molecules[11–13]. Extensive theoretical studies on the dynamics of AED exist for various systems, including halogen anions colliding with hydrogen[14], the creation of hydroxyl molecules from $O^-$ and hydrogen[15], the collisions of Li + $H^-$ [16], and the fundamental reaction H + $H^-$ [9,17]. In contrast, detailed experimental studies are limited to only few examples exploring reaction paths to the destruction of astrochemically relevant anions[2,7,9,17].

Our work presents a detailed experimental investigation on the AED reaction dynamics between hydroxyl anions ($OH^-$) and a cloud of laser-cooled $^{85}$Rb atoms, in a hybrid atom-anion trap. The AED process in this system involving a closed-shell anion and a single active electron atom is characterized by the emergence of an intermediate dipole-bound complex. Unlike valence-bound anions where the electron is characterized by dense, localized and multiply occupied orbitals, the excess electron in a dipole-bound anion lies in a very diffuse, singly-occupied orbital[18–20]. Historically, dipole-bound anions went from just being a theoretical curiosity to becoming identified as important species in various chemical processes, e.g electron capture in neutral molecules[21,22], zwitterion chemistry which plays an important role in amino acids[23] and charge transfer processes[24,25]. In astrochemistry, dipole-bound anions have been invoked as important precursors to the formation of valence-bound anions[26], and as candidates for the explanation of diffuse interstellar bands[27,28]. For our system, the intermediate dipole-bound anion exhibits a stable ground state and a short-lived excited state, resulting in a vastly different dynamics of the AED reaction.

Over the last years, the study of controlled ion-neutral reactions has enabled insights into the collisional dynamics and the investigation of chemical phenomena at their most elementary level[29–32]. However, most work focused on cationic and neutral systems, leaving out important collisions and reactions involving negative ions[33]. Also, a comprehensive experimental study of the electronic state-dependence in controlled reactions is largely unexplored.

In our work, making use of state-of-the-art techniques for trapping of ions and atoms[31,34], we can precisely control the amount of excited rubidium, allowing us to explore the influence of the electronic state on the anion-neutral reaction dynamics. The observed experimental results are compared to predictions of the Langevin classical capture model[35] and ab initio calculations performed for the Rb–$OH^-$ system[36,37]. As we show, the Langevin model fails to explain the reaction dynamics for both the ground and excited state. In contrast, the ab initio calculations, including steric effects, yield good quantitative agreement with the observed reaction rate coefficients for the ground state and provide a qualitative interpretation for the dynamics involving the excited state.

## Results

**Theoretical description of associative electronic detachment.** The only energetically accessible loss channel for Rb in the ground state is the AED reaction $Rb(^2S) + OH^- \rightarrow RbOH + e^-$, as theoretically investigated in[36,37]. For excited Rb, there are additional loss channels, of which the AED channel dominates (see Supplementary Note 4). In order to understand the underlying mechanism involved in the measured loss processes, we apply a modified Langevin capture model which takes into account that the AED is energetically allowed only for a finite range of angles of approach. For the ground state complex, the crossing between the anionic and neutral the potential energy surface (PES) occurs only at short range and not at long or intermediate internuclear distances due to the weak binding of the dipolar intermediate complex[36,37]. The PESs of the ground and low-lying electronic excited states of the anion Rb–$OH^-$ and neutral Rb–OH collisional complex have been calculated using ab initio methods (for details see Methods and Supplementary Note 1). A 1D cut of the PES at an exemplary collisional angle of $\theta = 80°$ is shown in the left panel of Fig. 1a. The region for which the energy of the anion is larger than the energy of the neutral (when the anion PES intersects the neutral one) is defined as the autodetachment region (gray shaded area).

The ground state of the intermediate complex Rb–$OH^-$ (blue curve in Fig. 1a) is stable against autodetachment as its energy lies below the neutral one. It can be categorized as a dipole-bound state, despite its rather large detachment energy ($\approx 0.3$ eV)[38]. The autodetachment region can only be reached for a limited angular space in the repulsive part of the PES (see the two cases of $\theta = 80°$ and $0°$ in Fig. 1a, where the crossing is energetically inaccessible and accessible respectively). Thus, a much lower reaction rate than the upper bound given by the Langevin capture rate is to be expected.

More quantitatively, based on our ab initio calculations (see[36]), the reaction path for the AED reaction is shown in Fig. 1b. The reaction starts with the Rb + $OH^-$ reactants, with the excess electron occupying a valence-bound $\pi$-orbital. The shape of the HOMO changes drastically when the reaction proceeds to the formation of the Rb–$OH^-$ intermediate complex exhibiting the typical halo shape of a dipolar bound complex. The stability of a dipole-bound anion primarily depends on the dipole moment of the core. The dipole moment of the Rb–OH decreases with decreasing interatomic distance $R_{Rb}$ (as shown in the Supplementary Fig. 2). Thus, a crossing with the energy of the neutral state can only occur in the repulsive inner region of the PES (see Fig. 1a, right panel).

As the dipole moment increases with $\theta$, the anionic states become stabilized beyond a critical threshold value, which defines the angular space fraction $\rho$, in which the AED reaction occurs (see Methods for details). For $\theta < 20°$ (dashed blue levels shown for $\theta = 0°$), the energy crossing is found below the entrance channel threshold opening the AED channel (blue dot in Fig. 1a right panel). For $\theta \gtrsim 20°$ (solid blue levels shown in Fig. 1b at $R_c$ for $\theta = 80°$) the crossing becomes inaccessible. In this case, $\rho$ is rather small, leading to a large deviation from the capture theory. The steep rising potential in the repulsive region, leads to a reaction probability highly sensitive to the ab initio methods and, in particular, to the effective core potential used[36,37]. As shown in Table 1, we find a reaction rate to be a factor of ten smaller than the Langevin rate employing the best available effective core potential for Rb. The calculation for the Langevin capture rate is described in Supplementary Note 3.

The electronic states of the dipole-bound intermediate complex, that are embedded into the autodetachement region, correlate to the excited entrance channel $Rb(^2P) + OH^-$ (red curves in Fig. 1a).

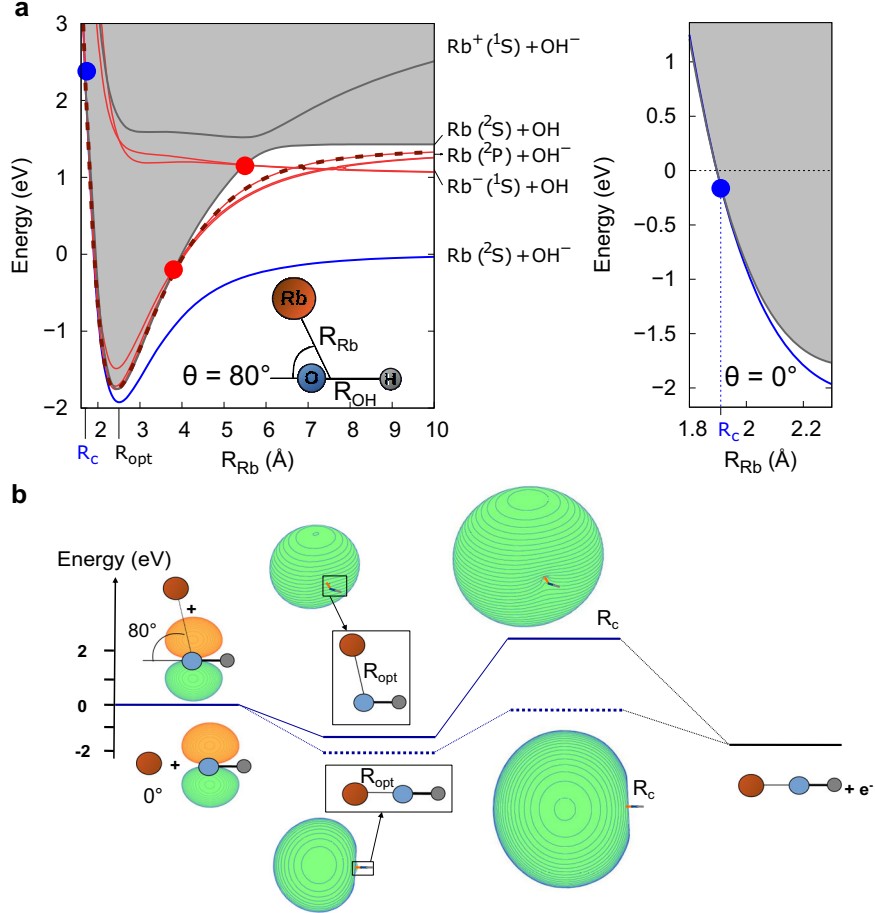

**Fig. 1 Description of the Rb–OH⁻ system. a** Potential energy surfaces as a function of the distance $R_{Rb}$ between the Rb atom and the center of mass of O–H (left panel). The angle $\theta$ between the OH axis and the Rb atom is chosen to be 80°. The diabatic crossing between the excited state RbOH⁻ complex (red curves) and its neutral counterpart RbOH (gray curve), is indicated by the two red dots. The crossing for the ground state RbOH⁻ complex occurs at the inner part of the potentials (blue dot). $R_{opt}$ stands for the optimized $R_{Rb}$ distance (minimum of the interaction well) and $R_c$ corresponds to the distance at which the detachment occurs. The crossing in the repulsive region between the anionic and neutral PES for $\theta = 0°$ is shown in the right panel. **b** Reaction path for the AED reaction between ground state Rb and OH⁻ shown for two different collisional angles: $\theta = 0°$ (dashed blue line) and $\theta = 80°$ (solid blue line). The zero corresponds to the energy of the Rb+OH⁻ entrance channel. The orbital corresponding to the excess electron (highest occupied molecular orbital, HOMO) is shown.

**Table 1 Comparison of results from the classical capture theory predictions (Langevin) and the ab initio calculations (modified capture model), to the experimentally obtained reaction rate coefficients (at 355 K).**

|  | $k_{GS}$ ($10^{-9}$ cm³ s⁻¹) | $k_{ES}$ ($10^{-9}$ cm³ s⁻¹) |
| --- | --- | --- |
| Langevin | 4.3 | 7.2 |
| Ab initio calculations | 0.42 | 7.5 |
| Experiment | 0.85(7) | 2.1(4) |

$k_{GS}$ and $k_{ES}$ are the ground state and excited state reaction rate coefficients, respectively.

Since all possible pathways lead to an energetically accessible crossing into the autodetachment region, which ultimately leads to AED, the total loss rate is expected to be close to the capture rate. Due to the highly diabatic nature of the PESs, the dynamics of the AED reaction can be primarily described as following a single diabatic PES (red dashed curve in Fig. 1a left panel). The excited state of the dipole-bound intermediate complex is short-lived and undergoes spontaneous autodetachment.

However, for increasing collisional angles, the dipole moment increases and the excited state stabilizes similar to the ground state (see Supplementary Fig. 2). In particular, above a critical collisional angle, here $\theta > 150°$, the crossing into the autodetachment region occurs at an energy higher than that of the entrance channel. This critical collisional angle defines the accessible angular space $\rho$, for AED with an excited state rubidium and is significantly larger than the ground state case. The calculated loss rate is obtained using appropriate long-range interactions and features of the PES (see Methods). It is found to be close to the Langevin rate (see Table 1), which is explained by the cancellation of additional long-range interaction terms and the reduced accessible angular space.

**Measurement of reaction rate coefficients**. A mass-selected ensemble of hydroxyl anions OH⁻ formed via electron attachment is loaded into an octupole radio-frequency wire trap, as schematically shown in Fig. 2a and described in detail in[39,40]. Multipole ion traps feature a large field-free region in the radial direction, thus reducing radio-frequency heating[41]. The hydroxyl ions occupy the vibrational ground state, as all higher vibrational states decay on a millisecond time scale. The kinetic temperature

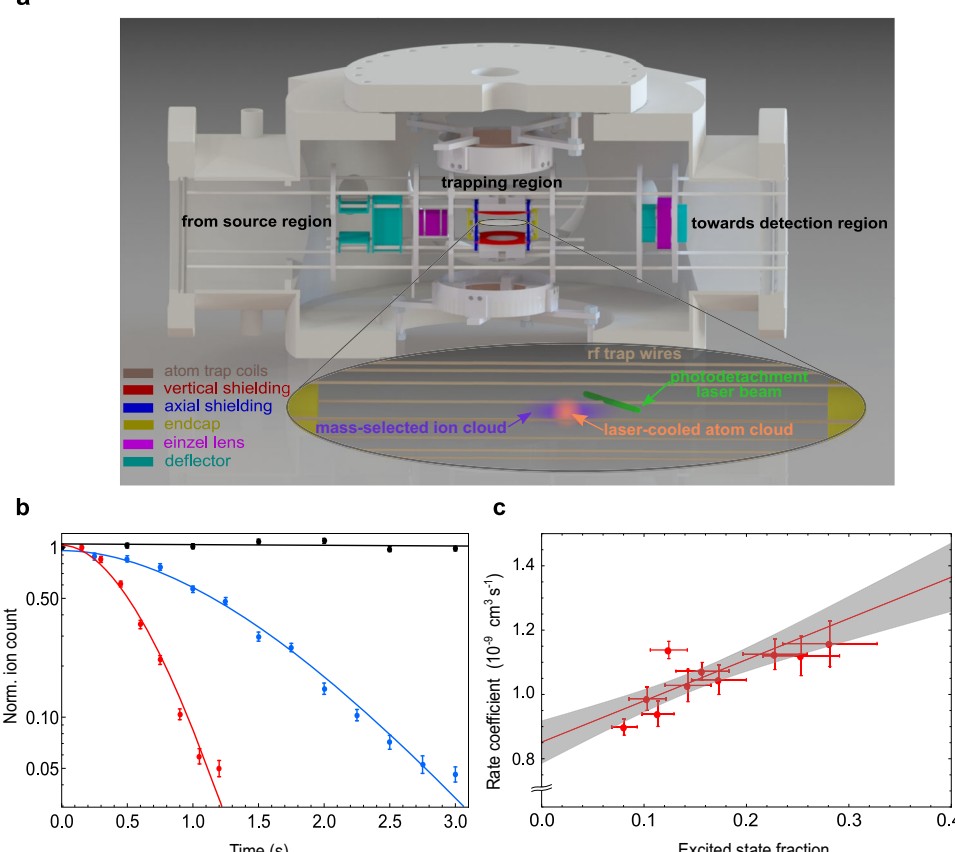

**Fig. 2 Hybrid atom-ion system. a** The experimental hybrid atom ion trap system. The OH⁻ ions (purple cloud) are created and loaded from the source chamber and trapped in an octupole radio-frequency (rf) wire trap. A far-threshold laser beam (green) is used to determine the ion density via photodetachment tomography. A laser-cooled cloud of ultracold Rb atoms (orange cloud) is overlapped with the ion cloud. As shown here, the spatial extent of the ion ensemble is significantly larger than that of the atomic cloud. After interaction with the laser or atoms, the time of flight of the ions are extracted onto the detector. **b** The detected normalized OH⁻ ion count after reaction with the laser-cooled rubidium atoms for excited state fractions of 0.10(2) and 0.28(5) (blue and red data points, respectively). The gray data points depict the ion losses without the presence of rubidium atoms. The ion losses are fitted by Eq. (1) (solid lines). The error bars represent the statistical errors. **c** Reaction rate coefficient as a function of the excited state fraction. The solid red line is a linear fit through the data points. The slope and intercept of the fit yields the reaction rate coefficient for the excited state and ground state rubidium interacting with OH⁻, respectively. The gray shaded area represents the 1-$\sigma$ range of statistical uncertainty.

is set to 355(10) K, via collisions with a pulse of helium buffer gas. The temperature of the ions is measured by mapping the ions' energy distribution to their time-of-flight (TOF) to the detector[40]. The ions' spatial distribution is mapped out by photodetachment tomography with a far-threshold laser[42].

Once the ions are trapped, they are overlapped with an ultracold cloud of rubidium atoms loaded into a dark spontaneous-force optical trap (dark-SPOT) configuration[43]. In the dark-SPOT version of a magneto-optical trap, a part of the repumping laser beam is spatially blocked. By changing the intensity of this laser beam, the fraction of atoms that are pumped back into the cooling cycle is controlled, thus providing control of the fraction of atoms in the excited (Rb($^2$P)) and ground state (Rb($^2$S)). After a given reaction time, the number of ions remaining are extracted onto a detector, thus yielding a loss rate dependent on the fraction of electronically excited rubidium in the ensemble.

The ion losses for two different excited state fraction of the atom ensemble, are shown in Fig. 2b. The evolution of the ion number in the ensemble, $N_I$, is expressed as:

$$N_I(t) = N_I^0 \cdot \exp\left(-k \int_0^t \Phi_{IA}(t')dt'\right) \cdot \exp(-k_{bgd}t) \quad (1)$$

where $N_I^0$ is the initial ion number, k is the reaction rate coefficient, $\Phi_{IA}$ is the spatial overlap between the ion and atom cloud (see Methods) and $k_{bgd}$ is the background ion loss rate without the presence of atoms. The ion losses are fitted by Eq. (1) yielding the reaction rate coefficient for the corresponding excited state fraction. By varying the amount of excited Rb in the atomic ensemble, a linear relationship between the reaction rate coefficients and excited state fractions is found as shown in Fig. 2c. From the intercept of a linear fit through the data points, the reaction rate coefficient for Rb in the ground state is obtained as $k_{GS} = 8.5(7) \times 10^{-10}$ cm$^3$ s$^{-1}$ with the corresponding statistical uncertainty. The slope yields the reaction rate coefficient for excited state as $k_{ES} = 2.1(4) \times 10^{-9}$ cm$^3$ s$^{-1}$. We estimate a systematic uncertainty of 40% and 60%, respectively, mainly due to the determination of the spatial overlap $\Phi_{IA}$ and parameters of the atom cloud.

## Discussion

The experimentally observed reaction rate coefficients are compared with the predictions of the classical capture theory and our ab initio calculations (modified capture model) as shown in Table 1.

The reaction rate for the ground state channel obtained from the experimental results shows the expected deviation from the Langevin prediction due to steric effects determining the stability of the intermediate dipolar complex, as previously outlined. The dipole moment drops below the critical threshold only for certain angles of approach for the anion-neutral collision, rendering a substantially reduced angular space where the autodetachment region is accessible. The experimental results confirm our understanding of the AED reaction dynamics, where for the first time, the dominant influence of a dipole-bound state as a critical reaction intermediate is revealed.

In other alkali hydroxide anions, the autodetachment region is predicted to generally lie above the energy of the entrance channel[38]. In such cases, depending on the energy gap between the anionic state and neutral state, diffuseness of the dipole-bound electron, and the reduced mass of the system, the AED rate is essentially determined by the presence of non-adiabatic coupling of the discrete states with the continuum[16,44].

In general, alkali atoms when interacting with non-metal or halogen anionic species, will form complexes with large dipole moment due to the large difference in the electronegativity. These collisional complexes will most likely support dipole-bound excited states. Our results, thus provide a framework for probing the influence of stable dipole-bound states for future studies on anion-neutral reactions.

As shown in Table 1 for the excited state channel, the experimentally determined reaction rate deviates significantly from the capture model as well as the ab initio calculations. This indicates the presence of additional stabilization mechanisms, like, e.g., the presence of longer lived metastable intermediate states in combination with non-adiabatic couplings, not accounted for in the current theoretical model.

In conclusion, we investigate the electronic quantum state-dependent AED rates in the Rb–OH⁻ system. Through control on the electronic configuration of the rubidium atom, the electronic state of the intermediate dipolar complex is altered and its influence on the collisional detachment process is revealed. The intermediate complex is a dipole-bound anion which exhibits a stable ground state but short-lived excited states. A unifying feature of the reaction dynamics for both the ground and excited state channels is the accessible angular space which governs the probability of the reaction to occur. For the ground state Rb interacting with OH⁻, the experimentally observed rate deviations from the capture model predictions are explained via the steric effects. Due to the high sensitivity of the reaction rate coefficient on the subtle details of the structure of the intermediate complex, our measurements provide a stringent test for different effective core potential models. However, for the excited state, the measured loss rate is significantly lower than the ab initio and capture model predictions. A deeper theoretical investigation of the excited states of the dipole-bound Rb–OH⁻ complex is needed to understand this discrepancy. Due to its similarity in the electronic structure of the intermediate complex to other alkali, alkali-earth hydroxides[38], and alkali with hydrated hydroxide[45], this work provides a general experimental framework to investigate state dependent alkali-anion reactions and opens new routes for the understanding of AED reactions.

## Methods

- Determination of spatial overlap, $\Phi_{IA}(t)$ With an ion spatial density distribution constant in time (determined via ions' time-of-flight distribution), the evolution of spatial overlap between the ion and atom cloud is governed by the time-evolution of the atom cloud (loading behavior) $n_A(t) = n_A^0 \left(1 - \exp(-t/\tau_A)\right)$, where $n_A^0$ is the peak atom density and $\tau_A$ is the loading time. The volume of the atom cloud governs the boundary of

the interaction region. The overlap $\Phi_{IA}(t)$ can be determined as:

$$\Phi_{IA}(t) = \int \bar{n}_I(x,y,z) \cdot n_A(x,y,z,t)\,dxdydz \qquad (2)$$

Here, $\bar{n}_I(x,y,z)$ is the unit-integral normalized OH⁻ density and $n_A(x,y,z,t)$ is the time-dependent atom density distribution. The ion density distribution is determined via photodetachment tomography[46]. The potential landscape arising from the geometry of an octupole wire trap results in the ion density distribution radially proportional to $r^{2n-2}$ (where $n=4$), and axially represented by a Gaussian profile. The atom cloud exhibits a Gaussian density profile determined via saturation absorption imaging[47,48]. The spatial extent of the ion cloud is much larger than that of the atomic cloud. The number of excited atoms are imaged by fluorescence imaging while the total number of atoms are imaged via saturation absorption imaging. The ratio of the two gives the excited state fraction of the atom cloud.

- Theoretical method: The rate coefficients corresponding to the reaction involving Rb(²P) and OH⁻ have been obtained using a modified capture model that includes features of the spin-orbit PES (see Supplementary Note 2), along with the following assumptions:

    - Since the rubidium atom is present in its Rb(²P₃/₂) fine state in the trap, the collision with OH⁻ can either proceed following the $6E_{1/2}$ or $5E_{1/2}$ spin-orbit PES of the Rb–OH⁻ complex. In the first case, the reaction proceeds on a potential that exhibits a barrier (see Supplementary Fig. 4), in the second case the upper limit of the cross section is taken to be the capture cross section $\sigma_Q$ obtained with the following long-range potential: $V(R) = \frac{-\alpha}{2R^4} + \frac{-Q}{2R^3} + \epsilon\left(\frac{b}{R}\right)^2$ where $\alpha$ and $Q$ are the static polarizability (870 a.u.) and quadrupole moment (26 a.u.) of Rb in its ²P state.
    - The detachment of the excess electron is assumed to be instantaneous when the autodetachment region is reached (sudden approximation).
    - The transition probability between adiabatic potentials have been calculated using the Landau–Zener formula[49].
    - Electronic detachment can only be avoided if the collision takes place along the $2E_{1/2}$ PES (first excited states of the Rb–OH⁻ complex) within the angular space $\rho = \frac{1}{2}(1 - cos(\theta_{max}))$[36], where $\theta_{max}$ is the collision angle for which it crosses the neutral curve above the energy in the entrance channel. We found $\theta_{max} \approx 153°$ (see Supplementary Fig. 3).
    - The electronic to kinetic energy transfer reaction rate is expected to be very small. This reaction has therefore been neglected (see Supplementary Note 4).

The total loss cross section from the Rb(²P₃/₂) + OH⁻(²Σ⁺) entrance channel is given by a sum of the loss from the $5E_{1/2}$ and $6E_{1/2}$ channels:

$$\sigma_{loss}(E_c) = \frac{1}{2}\sigma^{6E_{1/2}}(E_c) + \frac{1}{2}\sigma^{5E_{1/2}}(E_c) \qquad (3)$$

The first and second term are obtained using the following expressions:

$$\sigma^{6E_{1/2}}(E_c) = \left(1 - P_{NR}(E_c)(1-\rho)\right)\sigma_Q(E_c) \qquad (4)$$

$$\sigma^{5E_{1/2}}(E_c) = \left(1 - P_{NR}(E_c)(1-\rho)\right)\sigma_B(E_c) \qquad (5)$$

where $(1-\rho)$ corresponds to the angular space where the crossing into the auto-detachment region is avoided, $P_{NR}$ is the Landau–Zener probability to exit through the non-reactive channels Rb(²P₃/₂) + OH⁻(¹Σ⁺) or Rb(²P₁/₂) + OH⁻(¹Σ⁺) (thus $1-P_{NR}$ is the probability to exit via the charge transfer channel), $\sigma_Q$ is the capture cross section and $\sigma_B = \pi R_B^2(1 - U_B/E_c)$ is a classical cross section. The latter is given in terms of the largest impact parameters $b_{max}$ for which the potential barrier is less than the collision energy $E_c$. $R_B = 34$ a.u. and $U_B = 4.5 \times 10^{-3}$ are the position and height of the potential barrier, respectively (see Supplementary Fig. 4). The rate constant is then obtained by averaging over a Maxwell–Boltzmann distribution:

$$k_{loss} = \sqrt{\frac{8}{\pi\mu(k_b T)^{3/2}}} \int \sigma_{loss}(E_c)E_c\, e^{\frac{-E_c}{k_b T}}\,dE_c \qquad (6)$$

where $\mu$ is the reduced mass of Rb–OH⁻ system.

Owing to the highly diabatic nature of the PES (i.e small Landau–Zener adiabatic transition probability, (see Supplementary Note 4)), the probability to exit through the charge transfer channel, $P_{CT}$, is very small, around 1.5% for the relevant collision energies. Therefore $P_{NR} \approx 98.5\%$ and the entire dynamics is controlled by $\rho$. For $\rho = 1$, $\sigma_{loss} = \frac{1}{2}(\sigma_B + \sigma_Q)$ which leads to a capture case where all collisions that overcome the centrifugal and potential barrier, resulting in AED. For $\rho = 0$, the $2E_{1/2}$ state is stable against autodetachment (its energy lies below the neutral for all collision angles $\theta$). The total loss is then given by $\sigma_{loss} = \frac{1}{2}(1 - P_{NR})(\sigma_B + \sigma_Q)$. Hence, associative detachment can only occur following the adiabatic states from the entrance channel for which the Landau–Zener probability is $(1 - P_{NR}) = 1.5 \times 10^{-2}$. With the factor of 0.5, the total loss becomes $\sigma_{loss} = 7.5 \times 10^{-3}(\sigma_B + \sigma_Q)$.

## Data availability

The data that support the findings of this study are available from the authors upon request.

## Code availability

The codes and analysis files that support the findings of this study are available from the authors upon request.

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

## Acknowledgements

The authors would like to acknowledge Dr. Bastian Höltkemeier for his role in the realization of the experimental setup. This work is supported by the Austrian Science Fund (FWF) through Project No. I3159-N36 and Deutsche Forschungsgemeinschaft (DFG) under Project No. WE/2661/14-1. S.Z.H. acknowledges the support from IMPRS-QD fellowship and HGSFP. M.K. is grateful to the BMBF project MeSoX (Project No. 05K19GUE) for financial support.

## Author contributions

R.W. and M.W. conceived the project, S.Z.H., J.T. and H.L.C. performed experimental design and implementation, S.Z.H., M.N. and E.E. analyzed the experimental results, M.K. performed the theoretical investigations, S.Z.H. and M.K. wrote the manuscript. All authors contributed to the editing and revision of the manuscript.

## Funding

## Competing interests

The authors declare no competing interests.
