## [Peer Review File · Nature Communications]

REVIEWER COMMENTS

Reviewer #1 (Remarks to the Author):

The article by Hassan et al. describes a combined experimental and theoretical study of associative electron detachment (AED) reactions between hydroxyl anions and cold Rb atoms in an ion-atom hybrid trapping experiment. Rate constants for AED have been measured for collisions with Rb in its ground and electronically excited state. The experimental rate constants have been compared with theoretical predictions based on capture theory and extensive ab-initio calculations. The present work provided new insights into the state-dependence and stereodynamics of AED reactions and illustrated the importance of dipole-bound states as reaction intermediates. The authors should be congratulated on accomplishing this set of challenging experiments and calculations. The data and their analysis appear sound, the paper is generally well written and the results will certainly be of interest to a broad audience in both chemistry and physics. I can therefore recommend acceptance of this manuscript after the authors have considered and addressed the following points:

- The structure of the paper is somewhat unorthodox; the experimental setup and measurements are only described after the presentation of both the theoretical and experimental results. It would be more natural to place the section "Measurement of reaction rate coefficients" at the very beginning of the Results section.

~~[1]~~ The readability of the paper could be further improved by briefly summarising the salient ideas of the theoretical model for calculating the rate constants already in the main text.

- The experiments employ a dark-SPOT trap for the Rb atoms, in which the repumper beam is blocked in the centre. Thus, the excited-state Rb atoms are localised at the extremities of the atom cloud and their spatial distribution is strongly inhomogeneous. Has this effect been accounted for in the calculation of the overlap between the ions and atoms, and how does it impact the accuracy of the rate constants determined for the excited state of Rb? In this context, the authors should also explain in more detail how they defined and determined the excited-state fractions in Figs. 2b and c.

- An important element of the analysis is the assumption that successful AED can only occur for different orientation angles of the reaction partners. However, as the dipole moment of the complex seems to strongly depend on the orientation angle, the potential is strongly anisotropic (see also Fig. 3 of the SI). One thus wonders about the importance of "steering forces" during the reaction and how they impact the reaction rates. A short discussion on this issue would be warranted.~~[1]~~

- The temperature of the Rb atoms in the MOT was not indicated, but it will be much smaller than the 355 K quoted for the ions. Does any appreciable sympathetic cooling of the hydroxyl anions by the Rb atoms occur in these experiments, and if so, what is its relevance for the present results ?

- p.9, last line of Methods section: how does the pre-factor $7.5e-3$ arise

Reviewer #2 (Remarks to the Author):

Review of the Manuscript NCOMMS-21-28192-T “Associative detachment in anion-atom reactions involving a dipole-bound electron” by Saba Zia Hassan, Jonas Tauch, Milaim Kas, Markus Nötzold, Henry López Carrera, Eric S. Endres, Roland Wester, and Matthias

Weidemüller, submitted to Nature Communications. In this manuscript, the authors investigate the interaction of OH⁻ with a cloud of ultracold Rb atoms in a hybrid atom-ion trap, and by measuring the loss rate of OH⁻, infer a rate coefficient for the associative detachment reaction. They explore the reaction for Rb in its ground state and in its first excited state, and find that the deviation of experimental rate coefficient from the theoretical prediction can be explained through steric effects for ground state Rb atoms, but not for excited Rb atoms. The authors make the point that because associative electronic detachment (AED) of anions is a general process playing an important role in the formation of a range of molecules and in a variety of environments/conditions, their study paves the way to understanding AED in general.

Overall, the manuscript is well written (including the supplementary document), and the results seem solid. The paper is a continuation of the work on this system by the theoretical and experimental groups, extending their previous investigations to excited Rb atoms with OH⁻ (e.g., see Refs. [34,36,37]). They extended the theoretical work to include the experimental determination of the rate coefficient as a function of the excited Rb atoms' population and extrapolated to zero population to extract the ground state rate. They also extend their previous analysis of the angle of approach dependence in [34] and use this information to obtain a modified/corrected Langevin-like rate coefficient.

In my opinion, the reason this manuscript could be considered is that it includes experimental determinations of the reaction rates. While this experimental technique might be general enough is made in the manuscript, the apparatus is using ultracold Rb atoms. Clearly, the experiment could employ a few other atomic species that can be cooled to ultralow temperatures, but it seems a bit

restrictive. Also, other anions could be explored, but the current approach is not yet ready for them. In that sense, it points to a possible “general” enough interest still far from realization. The theoretical extension to other anions has already been discussed by the authors in [36] and [43]. While I am conflicted about the real impact of these results on the field, I would recommend publication in Nat. Comm. after the authors consider the issues listed below.

First, in the abstract and introduction, the authors used the expression “chemical networks”; it is not clear to me what is meant by this. I would suggest using something like “chemical systems”. Also in the abstract, the use of “configuration of nuclei” could be understood (like I initially did) to be linked to the spin of the nuclei, not the physical location; I would suggest to use “location of the nuclei” instead. Again in the abstract, the last claim “electronic state dependent dynamics in AEDs previously unexplored” is not entirely correct, since the authors did such studies in previous publications [34,36].

In the introduction, the authors could refer to the initial theoretical work on Rb and OH⁻ in [35], which identify many of the issues later covered in [34] and [36]. Actually, [35] is mentioned in passing only the first sentence of section 2. While I understand the urge of citing one’s own work, and even if the older [35] did not study excited state and used a different electronic configuration basis set, the angle-dependent reaction rate was identified as well as the role of the J-distribution of OH⁻. Similarly, a general review chapter by Cote on atom-ion hybrid systems could be mentioned as well as the experimental result on charge transfer by Smith et al. (see reference at the end). I also note that zwitterion chemistry (second paragraph of introduction), while well known to most chemists, is not common knowledge to physicist: maybe adding something like “zwitterion chemistry which plays an important role in amino acids”.

In the “Results and Discussion”, first section on “Theoretical description of AED”, on p.4, bottom of 3rd paragraph, “employing” is repeated. In the next section “Measurement of reaction rate coefficients”, there is a description of the temperature of the OH⁻ and that they are thermalized to 355 K, via collisions with a pulse of helium buffer gas. Could the authors comment on the populations of OH⁻ (could also be in the discussion section instead). In fact, as discussed in [34] and [35], the J-distribution plays an important role: while it seems to be taken into account in (in section 2 of Supplemental document), it has not been made clear. Actually, looking at Ref.[34], the J=1 and 2 seem to contribute more to the process than J=0: this should be discussed. In fact, the effect of the J-distribution could affect the effective position of the barriers for excited state. Similarly, while the v=0 is probably the most populated state of OH⁻ (as mentioned in [34]), the OH⁻ is produced by discharge (mentioned in [34] but not here, at least it was not obvious), it was pointed out in [35] that higher v’s would impact the rate significantly: even a small population could explain some of the discrepancies for the higher rate for the ground state Rb (probably not, but should be commented on).

Another point to be considered is the de-excitation of Rb atoms as they collide with OH-. This was studied in [29] and played an important role in the reaction rate. While the temperatures are much higher here and the collisions much faster (so that de-excitation might not play an important role), they could help explaining the lower rate for excited Rb atoms.

Looking at Fig.2 c), I am wondering why is the rate coefficient depending on the fraction of excited Rb ? I can understand the rate, which depends on the density and thus the fraction of Rb, but I thought a rate coefficient should be independent of that fraction (basically, you determine the rate coefficient assuming a single Rb interaction with a single OH- (a velocity times a cross section)). Is the quantity plotted in Fig.2 c) the rate ? Also, why the “peak” near fraction = 0.13 ? Is there anything special in the apparatus to explain this point that is outside the uncertainty band ? Some more discussion on those issues would make sense. An interesting question is how dependent are the rates from the fine state of excited Rb atoms. Does it matter at all (maybe not in the present experiment due to the collision energy), but it clearly would affect curve crossings. A mention of it would also be a good idea.

Finally, in the Supplementary documentation, as I mentioned above, the J-distribution is implicitly taken into account. It might be a good place to talk more about it (and possibly the v-distribution).

So, overall, this is a solid piece of work, showing incremental theoretical work. The main contribution, in my opinion, is the experimental measurement and determination of the rate coefficient for both ground state and excited state Rb. I would recommend publication after the authors consider the points above.

Additional references:

R. Cote, “Ultracold Hybrid Atom-Ion systems”, in Ad. in AMO Physics, Vo. 65, pp. 67-126 (2016).

W.W Smith, D.S Goodman, I. Sivarajah, J.E Wells, S. Banerjee, R. Cote, H.H. Michels, J.A. Montgomery, F.A. Narducci “Experiments with an ion-neutral hybrid trap: cold charge-exchange collisions”, Applied Physics B 114, 75-80 (2014).

Reviewer #3 (Remarks to the Author):

This is a very interesting manuscript describing a combined experimental and theoretical study of associative detachment (AED) in OH- Rb collisions. Authors demonstrate the critical importance of dipole bound intermediate complex. The Rb-OH- complex where Rb is in the ground electronic state is stabilized against AED and the autodetachment can only occur at near linear configuration severely limiting the accessible phase space for the reaction. The picture is dramatically different for the electronically excited Rb. In this case the phase space available for the AED is almost as large as expected for the Langevin limited process.

Authors convincingly demonstrate in their experiment that the model provides a qualitative description for the AED process. Moreover, intriguingly the measured AED rate is more than a factor of 3 lower as compared with the one predicted by ab-initio calculations. This, as authors discuss, indicates a different stabilization mechanism in play that currently is not taken into account.

The manuscript is a very important contribution to the modern molecular physics and provides a very clear description and experimental demonstration of a novel dipole driven stabilization mechanism that might be important in many similar systems. The manuscript is very well written and I strongly support publication. My only request to authors is to estimate if the weak magnetic field gradient of the MOT coils influences the motion dynamics of co-trapped anions.

REVIEWER COMMENTS

Reviewer #1 (Remarks to the Author):

The article by Hassan et al. describes a combined experimental and theoretical study of associative electron detachment (AED) reactions between hydroxyl anions and cold Rb atoms in an ion-atom hybrid trapping experiment. Rate constants for AED have been measured for collisions with Rb in its ground and electronically excited state. The experimental rate constants have been compared with theoretical predictions based on capture theory and extensive ab-initio calculations. The present work provided new insights into the state-dependence and stereodynamics of AED reactions and illustrated the importance of dipole-bound states as reaction intermediates. The authors should be congratulated on accomplishing this set of challenging experiments and calculations. The data and their analysis appear sound, the paper is generally well written and the results will certainly be of interest to a broad audience in both chemistry and physics. I can therefore recommend acceptance of this manuscript after the authors have considered and addressed the following points:

We are very thankful to the referee for the positive feedback and constructive comments on our manuscript. The comments are individually addressed as follows.

- The structure of the paper is somewhat unorthodox; the experimental setup and measurements are only described after the presentation of both the theoretical and experimental results. It would be more natural to place the section “Measurement of reaction rate coefficients” at the very beginning of the Results section.

Our intention was to already introduce the reader to existing theoretical qualitative and quantitative picture of the Rb-OH⁻ system, so they have a fair idea of what one might expect from the experiment. Hence the theory is followed by the experiment to avoid switching between the theoretical and experimental results. This is also the reason why in Table 1 the order of , Langevin, ab-initio and experimental results, is followed. However, to make the structure more coherent, we now move Table 1 after the section ‘Measurement of reaction rate coefficients’, and the caption of Table 1 is also modified.

- The readability of the paper could be further improved by briefly summarising the salient ideas of the theoretical model for calculating the rate constants already in the main text.

We added some sentences in the first paragraph of the section ‘Theoretical description of AED’ briefly explaining the idea of a modified Langevin model.

- The experiments employ a dark-SPOT trap for the Rb atoms, in which the repumper beam is blocked in the centre. Thus, the excited-state Rb atoms are localised at the extremities of the atom cloud and their spatial distribution is strongly inhomogeneous. Has this effect been accounted for in the calculation of the overlap between the ions and atoms, and how does it impact the accuracy of the rate constants determined for the excited state of Rb ?

The core of the repumper beam is blocked with an opaque spot, such that a hollow sphere of repumping light surrounds the atom cloud. The spot used here is much larger than the actual size of the atom cloud, such that the amount of atoms present in the ‘halo’ (atoms in the course of being trapped) is negligible and do not impact the accuracy of the rate constants. We also verify this by mapping out the atom cloud’s spatial distribution via saturation absorption imaging.

The ion cloud is significantly larger than the atom cloud, such that the atomic ensemble is completely embedded in the ion cloud. Thus, in the region of interest i.e. the expanse of the ion cloud, the atom cloud is homogeneous.

We now include an additional sentence in the caption of Figure 2, “As shown here, the spatial extent of the ion ensemble is significantly larger than that of the atomic cloud.”

In this context, the authors should also explain in more detail how they defined and determined the excited-state fractions in Figs. 2b and c.

An additional sentence in the Methods is included explicitly to explain this: “The spatial extent of the ion cloud is much larger than that of the atomic cloud. The number of excited atoms are imaged by fluorescence imaging while the total number of atoms are imaged via saturation absorption imaging. The ratio of the two gives the excited state fraction of the atom cloud.”

- An important element of the analysis is the assumption that successful AED can only occur for different orientation angles of the reaction partners. However, as the dipole moment of the complex seems to strongly depend on the orientation angle, the potential is strongly anisotropic (see also Fig. 3 of the SI). One thus wonders about the importance of “steering forces” during the reaction and how they impact the reaction rates. A short discussion on this issue would be warranted.

Including the steering force will definitely increase the loss from both the ground state and excited state channel. Indeed, the “head-on” collision Rb-O-H (corresponding to a collision angles of $\theta = 0^\circ$), leads to an increased probability of electronic detachment. For lower collision energies, the effect will start to be important and should be taken into account. In fact, the steering effect might partially explain the deviation of the measured rate coefficient from the predicted one. The effect is now discussed in section 3 of the Supplementary material.

- The temperature of the Rb atoms in the MOT was not indicated, but it will be much smaller than the 355 K quoted for the ions. Does any appreciable sympathetic cooling of the hydroxyl anions by the Rb atoms occur in these experiments, and if so, what is its relevance for the present results?

In the case of perfect overlap between the two clouds, there is no appreciable sympathetic cooling of the ion. This was also verified by mapping the ion distribution via photodetachment tomography with a far-threshold laser beam, for different interaction time with the MOT. The distribution did not significantly change for different interaction times. A high reaction rate observed actually results in the loss of anions even before they can be significantly sympathetically cooled via collisions with Rb atoms at $\sim 200\mu\text{K}$. By mapping the ions’ temperature to their time of flight to the detector, it was also observed that the change in the ion temperature is negligible. We comment about it in the Supplementary material (Section 3).

- p.9, last line of Methods section: how does the pre-factor $7.5e-3$ arise?

For $\rho=0$, the first excited state of the RbOH⁻ complex is stable against autodetachment. The total loss is then given by $0.5(1 - P_{NR})(\sigma_Q + \sigma_B)$. Hence, associative detachment can only occur following the adiabatic states from the entrance channel for which the Landau-Zener probability is $(1 - P_{NR}) = 1.5 \times 10^{-2}$. With the factor 0.5, the total loss becomes $7.5 \times 10^{-3}(\sigma_Q + \sigma_B)$. This explanation is also added in the Methods section now for making this part more clear.

Reviewer #2 (Remarks to the Author):

Review of the Manuscript NCOMMS-21-28192-T “Associative detachment in anion-atom reactions involving a dipole-bound electron” by Saba Zia Hassan, Jonas Tauch, Milaim Kas, Markus Nötzold, Henry López Carrera, Eric S. Endres, Roland Wester, and Matthias

Weidemüller, submitted to Nature Communications. In this manuscript, the authors investigate the interaction of OH⁻ with a cloud of ultracold Rb atoms in a hybrid atom-ion trap, and by measuring the loss rate of OH⁻, infer a rate coefficient for the associative detachment reaction. They explore the reaction for Rb in its ground state and in its first excited state, and find that the deviation of experimental rate coefficient from the theoretical prediction can be explained through steric effects for ground state Rb atoms, but not for excited Rb atoms. The authors make the point that because associative electronic detachment (AED) of anions is a general process playing an important role in the formation of a range of molecules and in a variety of environments/conditions, their study paves the way to understanding AED in general.

Overall, the manuscript is well written (including the supplementary document), and the results seem solid. The paper is a continuation of the work on this system by the theoretical and experimental groups, extending their previous investigations to excited Rb atoms with OH⁻ (e.g., see Refs. [34,36,37]). They extended the theoretical work to include the experimental determination of the rate coefficient as a function of the excited Rb atoms' population and extrapolated to zero population to extract the ground state rate. They also extend their previous analysis of the angle of approach dependence in [34] and use this information to obtain a modified/corrected Langevin-like rate coefficient.

In my opinion, the reason this manuscript could be considered is that it includes experimental determinations of the reaction rates. While this experimental technique might be general enough is made in the manuscript, the apparatus is using ultracold Rb atoms. Clearly, the experiment could employ a few other atomic species that can be cooled to ultralow temperatures, but it seems a bit restrictive. Also, other anions could be explored, but the current approach is not yet ready for them. In that sense, it points to a possible “general” enough interest still far from realization. The theoretical extension to other anions has already been discussed by the authors in [36] and [43]. While I am conflicted about the real impact of these results on the field, I would recommend publication in Nat. Comm. after the authors consider the issues listed below.

We extend our thanks to the referee for the positive feedback and the valuable critique of the manuscript. We are happy to address the comments in the following section.

First, in the abstract and introduction, the authors used the expression “chemical networks”; it is not clear to me what is meant by this. I would suggest using something like “chemical systems”.

We were referring to networks of chemical reactions. We believe that the term “chemical systems” may also be prone to misunderstanding. We suggest the use of “chemical reaction network” instead.

Also in the abstract, the use of “configuration of nuclei” could be understood (like I initially did) to be linked to the spin of the nuclei, not the physical location; I would suggest to use “location of the nuclei” instead.

This phrase ‘configuration of nuclei’ is now changed to ‘molecular geometry’ for clarity.

Again in the abstract, the last claim “electronic state dependent dynamics in AEDs previously unexplored” is not entirely correct, since the authors did such studies in previous publications [34,36].

The term “previously unexplored” has been ommited.

In the introduction, the authors could refer to the initial theoretical work on Rb and OH⁻ in [35], which identify many of the issues later covered in [34] and [36]. Actually, [35] is mentioned in passing only the first sentence of section 2. While I understand the urge of citing one's own work, and even if the older [35] did not study excited state and used a different electronic configuration basis set, the angle-dependent reaction rate was identified as well as the role of the J-distribution of OH⁻. Similarly, a general review chapter by Cote on atom-ion hybrid systems could mentioned as well as the experimental result on charge

transfer by Smith et al. (see reference at the end). I also note that zwitterion chemistry (second paragraph of introduction), while well known to most chemists, is not common knowledge to physicist: maybe adding something like “zwitterion chemistry which plays an important role in amino acids”.

We thank the referee for the comment and we now include the suggested references in the introduction as ref [29, 30]. An additional sentence in the first paragraph of the section ‘Theoretical investigation of AED’ is added to highlight the role of previous investigations in now ref. 37 (earlier 35). The works, in now ref [36,38], are more regularly cited and addressed, as the theory explains the experimental results observed. The authors duly acknowledge the role and importance of the work in [37] in providing a basis on later theoretical work adding to the understanding of Rb-OH- system and the reference is now included in introduction.

The sentence on zwitterion chemistry now includes the suggested additional phrase.

In the “Results and Discussion”, first section on “Theoretical description of AED”, on p.4, bottom of 3rd paragraph, “employing” is repeated.

This error has been corrected.

In the next section “Measurement of reaction rate coefficients”, there is a description of the temperature of the OH- and that they are thermalized to 355 K, via collisions with a pulse of helium buffer gas. Could the authors comment on the populations of OH- (could also be in the discussion section instead). In fact, as discussed in [34] and [35], the J-distribution plays an important role: while it seems to be taken into account in (in section 2 of Supplemental document), it has not been made clear. Actually, looking at Ref. [34], the J=1 and 2 seem to contribute more to the process than J=0: this should be discussed. In fact, the effect of the J-distribution could affect the effective position of the barriers for excited state. Similarly, while the v=0 is probably the most populated state of OH- (as mentioned in [34]), the OH- is produced by discharge (mentioned in [34] but not here, at least it was not obvious), it was pointed out in [35] that higher v’s would impact the rate significantly: even a small population could explain some of the discrepancies for the higher rate for the ground state Rb (probably not, but should be commented on).

Buffer gas cooling and thermalization is a very commonly used technique with traps mounted on cryostats that is discussed in detail in the references provided in the text (ref. 41 and 42). In previous work, it was found that collisions between OH- and He, lead to fast thermalization of the rotational levels of the ion ensemble, which especially at higher temperature corresponds to the external temperature of the ions. OH- is formed via electron attachment not via impact ionization like cations, which yields a very different energy distribution in the produced ions. Using the dipole moment of OH-, we find that the lifetime of v=1 state, before radiative decay is on the order of couple of milliseconds, which in comparison to the complete timescale of the experiment (~10 seconds), makes the fraction of ions in the excited vibrational states, negligible.

A comment about this is now included in the Supplementary material (Section 3).

Another point to be considered is the de-excitation of Rb atoms as they collide with OH-. This was studied in [29] and played an important role in the reaction rate. While the temperatures are much higher here and the collisions much faster (so that de-excitation might not play an important role), they could help explaining the lower rate for excited Rb atoms.

As the referee suggested, the temperature or collision energy of the system is too ‘high’ for this effect to play a role in influencing the reaction rates. This effect also cannot explain the factor 3 between calculated and measured rates.

Looking at Fig.2 c), I am wondering why is the rate coefficient depending on the fraction of excited Rb ? I can understand the rate, which depends on the density and thus the fraction of Rb, but I thought a rate coefficient should be independent of that fraction (basically, you determine the rate coefficient assuming a single Rb interaction with a single OH- (a velocity times a cross section). Is the quantity plotted in Fig.2 c) the rate ?

In the measurement we have an ensemble of atoms and ions, not looking at single collision events between single particles but averaged ensembles. By excited state fraction we mean that, from the ensemble of atoms, a fraction is in the excited state while the rest in ground state. And the two states individually contribute to the total reactivity of the Rb-OH-. In the picture of a single atom-ion collision, this would correspond to the probability of finding the atom in its excited state. Here, the quantity plotted in Fig.2c) is the Reaction-rate coefficient as a function of excited state fraction of the atomic ensemble.

Also, why the “peak” near fraction = 0.13? Is there anything special in the apparatus to explain this point that is outside the uncertainty band? Some more discussion on those issues would make sense.

This outlier point pertains to normal experimental noise. All the error bands are estimated via normal error propagation, which means that points don't have to necessarily lie in this region. As this is the standard procedure for experiment and error calculations, additional discussion was not deemed necessary, as it does not significantly alter the result.

An interesting question is how dependent are the rates from the fine state of excited Rb atoms. Does it matter at all (maybe not in the present experiment due to the collision energy), but it clearly would affect curve crossings. A mention of it would also be a good idea.

The PESs, from which the Landau-Zener transition probabilities and the position of the crossing point have been extracted for the dynamics, include the fine structure of Rb (see Figure 1 in Supplementary). They correspond to different dissociation channels that correlates to different molecular (spin-orbit) PES. Hence, leading to different "path" toward detachment or CT channels.

Finally, in the Supplementary documentation, as I mentioned above, the J-distribution is implicitly taken into account. It might be a good place to talk more about it (and possibly the v-distribution).

Following the referee's suggestion, an additional section that briefly discussed several issues raised by the referee has been added to the Supplementary material.

Reviewer #3 (Remarks to the Author):

This is a very interesting manuscript describing a combined experimental and theoretical study of associative detachment (AED) in OH- Rb collisions. Authors demonstrate the critical importance of dipole bound intermediate complex. The Rb-OH- complex where Rb is in the ground electronic state is stabilized against AED and the autodetachment can only occur at near linear configuration severely limiting the accessible phase space for the reaction. The picture is dramatically different for the electronically excited Rb. In this case the phase space available for the AED is almost as large as expected for the Langevin limited process. Authors convincingly demonstrate in their experiment that the model provides a qualitative description for the AED process. Moreover, intriguingly the measured AED rate is more than a factor of 3 lower as compared with the one predicted by ab-initio calculations. This, as authors discuss, indicates a different stabilization mechanism in play that currently is not taken into account.

The manuscript is a very important contribution to the modern molecular physics and provides a very clear description and experimental demonstration of a novel dipole driven stabilization mechanism that might be important in many similar systems. The manuscript is very well written and I strongly support publication.

We sincerely thank the referee for the positive feedback on our work and his support for publication.

My only request to authors is to estimate if the weak magnetic field gradient of the MOT coils influences the motion dynamics of co-trapped anions.

The OH- anion, is a closed-shell anion (in singlet sigma state) with no free electron to couple to the external magnetic field. With a negligible magnetic moment, the influence of the magnetic field gradient created by the MOT coils is very weak to influence the trajectory of the ions. Lorentz force can also be ignored at the low velocities present in the system.

REVIEWERS' COMMENTS

Reviewer #1 (Remarks to the Author):

The authors have addressed all my points adequately. I can now recommend this article for publication.

Reviewer #2 (Remarks to the Author):

After reading the new versions of the manuscript and Supplementary documentation, together with the response of the actors, I am completely satisfied, and believe that the manuscript should be published as is.

Reviewer #3 (Remarks to the Author):

All of the referees concerned have been fully addressed.